# S-Crescendo: A Nested Transformer Weaving Framework for Scalable Nonlinear System in S-Domain Representation

Junlang Huang[1,*], Hao Chen[1,*], Li Luo[1,†], Yong Cai[1,†], Lexin Zhang[1], Tianhao Ma[1], Yitian Zhang[1], Zhong Guan[1,✉]

## Abstract

Simulation of high-order nonlinear system requires extensive computational resources, especially in modern VLSI backend design where bifurcation-induced instability and chaos-like transient behaviors pose challenges. We present S-Crescendo - a nested transformer weaving framework that synergizes S-domain with neural operators for scalable time-domain prediction in high-order nonlinear networks, alleviating the computational bottlenecks of conventional solvers via Newton-Raphson method. By leveraging the partial-fraction decomposition of an n-th order transfer function into first-order modal terms with repeated poles and residues, our method bypasses the conventional Jacobian matrix-based iterations and efficiently reduces computational complexity from cubic $O(n^3)$ to linear $O(n)$. The proposed architecture seamlessly integrates an S-domain encoder with an attention-based correction operator to simultaneously isolate dominant response and adaptively capture higher-order non-linearities. Validated on order-1 to order-10 networks, our method achieves up to 0.99 test-set $R^2$ accuracy against HSPICE golden waveforms and accelerates simulation by up to $18\times$, providing a scalable, physics-aware framework for high-dimensional nonlinear modeling.

## 1 Introduction

In recent years, deep learning technologies have made remarkable progress, with Transformer-based architectures demonstrating exceptional performance and significant advantages in fields such as natural language processing (NLP), computer vision, and time-series data modeling.[1] Transformer models, by effectively capturing complex relationships and long-range dependencies, offer a novel perspective for data-driven modeling. This technological advancement has inspired researchers to explore its potential applications in traditional engineering domains, especially in complex physical modeling and signal prediction[2][3].

Nonlinear system identification remains a core challenge across many domains, particularly under high-order dynamics, non-stationarity, and limited observability. Classical methods from control theory—such as Volterra series, Hammerstein–Wiener models, and grey-box approaches—typically decompose system behavior into a linear core and a nonlinear correction [4, 5, 6, 7]. However, their scalability and generalization degrade in high-dimensional parameter spaces [8]. These limitations are especially pronounced in modern integrated circuit design, where nonlinearities emerge not only from active devices but also from parasitic effects in passive interconnects. A canonical example is the "nonlinear driver + linear RC load" configuration, illustrated in Figure 1. As technology scales, interconnect parasitics exhibit strong dynamic nonlinearities due to proximity effects, process variation, and material inhomogeneity [9], complicating accurate modeling. To address this, we propose a neural operator framework that integrates Laplace-domain physical priors with data-driven

39th Conference on Neural Information Processing Systems (NeurIPS 2025).

adaptability. Demonstrated on RC current response tasks, this method extends naturally to systems governed by partial differential equations with nonlinear boundary conditions, offering a scalable and physically consistent approach to complex system identification.

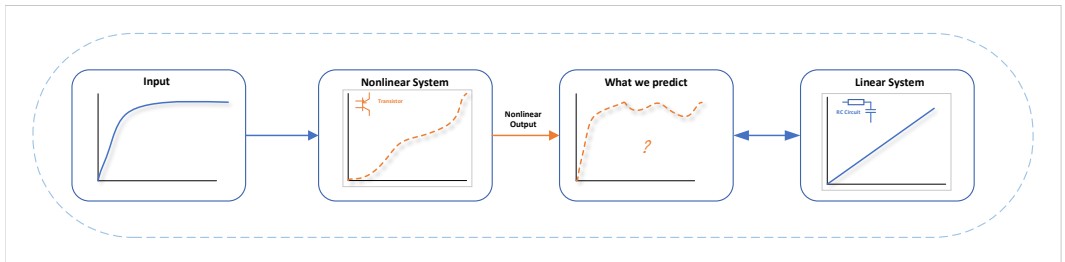

Figure 1: Given a known input signal, the task is to predict the nonlinear system's output before it feeds into the linear system. This intermediate signal, marked in red, is unknown and serves as the prediction target of our model.

In response, a growing body of research has applied deep learning to nonlinear system identification. Approaches based on recurrent networks[10], attention mechanisms[11], and neural operators[12] (e.g., DeepONet[13], FNO[14], and PINNs) have shown promise, particularly in low-dimensional settings or for PDE systems with known boundary conditions. Yet, these methods often require large datasets, lack robustness to structural variation, and suffer from poor physical interpretability[15][16]. More critically, few methods explicitly leverage the modal structure or transfer function representation intrinsic to many engineered systems. As a result, they remain constrained by either computational inefficiency (due to iterative solvers like Newton–Raphson) or limited generalization to systems with unseen topologies or higher-order dynamics[17][15][18][19].While deep learning has been explored for general system identification, its application to signal-line RC response modeling—a canonical high-order nonlinear problem—remains largely unexplored. Most existing methods still fall under three classical paradigms: Current Source Models (CSMs) [20, 21], Voltage Response Models (VRMs) [22], and Direct Waveform Prediction [23]. Each faces practical limitations: CSMs lack waveform fidelity due to fixed capacitance abstraction [24], VRMs suffer from high cost and solver-induced errors [25, 26], and direct fitting fails under sharp transitions due to overshoot and undershoot distortion.

Against this backdrop, Transformer-based methods offer a powerful new tool for nonlinear system identification in the S-domain. Transformer architectures excel at capturing complex, long-range dependencies and higher-order interactions, making them ideally suited to address the limitations of traditional RC-network models in time-domain response prediction [1].

Even in nominally high-order interconnects, only a handful of "dominant poles" govern signal behavior. Standard model order reduction (MOR) selects these modes, typically fewer than ten by energy or gain thresholds, and collapses a 5000th order RC network into a low-order surrogate with minimal frequency response error [27, 28]. Going beyond linear reduction, we encode each reduced first-order term into a learned latent embedding and apply an attention-driven correction operator to capture nonlinear driver-load effects. On 4th to 10th order surrogates, our method attains $R^2$ up to 0.99 against HSPICE waveforms, evidencing superior accuracy. By leveraging partial-fraction decomposition, we decouple topology from prediction-forecasting each modal response independently in the S-domain before summation, thus ensuring universality and structural agnosticism.We introduce a physics aware S-domain neural operator that seamlessly integrates with MOR pipelines, delivering a scalable, accurate, and efficient solution for nonlinear RC simulation, electromagnetic analysis, and signal-integrity evaluation without fixed driver or load models.

The proposed model adopts a two-stage architecture: a base module first predicts individual first-order responses from partial fraction decomposition and aggregates them into a baseline output. A subsequent compensation network then iteratively refines this output by learning residual corrections across model orders. This overall architecture is illustrated in Figure 2.

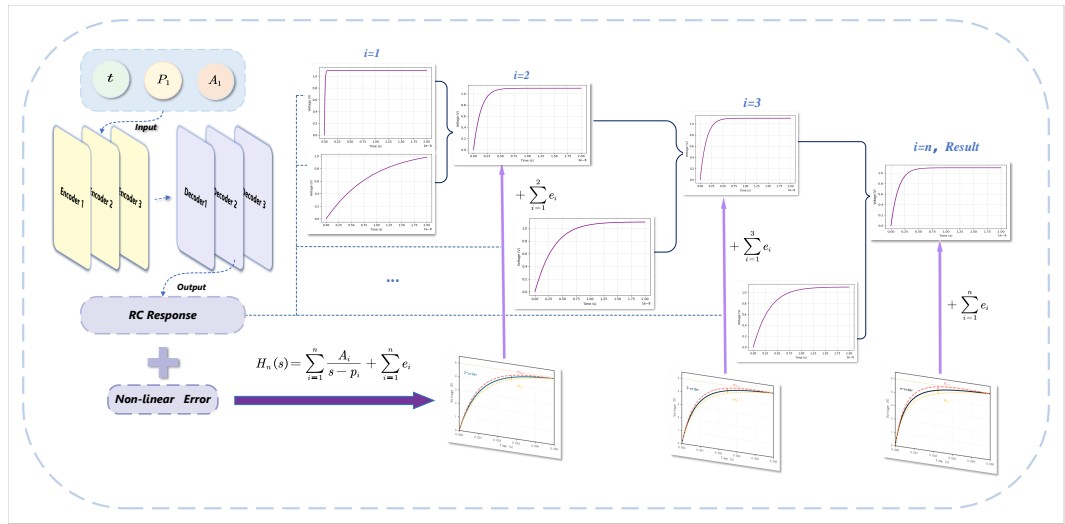

Figure 2: **Model Overview.** The model first constructs a baseline prediction by summing the first-order responses of pole-residue pairs. For each mode $i$, the residual module $e_i$ is trained using the current and previous poles, residues, and time information to correct the accumulated error. Residuals are added iteratively to refine the final prediction.

## 2 Theoretical Background

### 2.1 *Decomposition Theorem for High-Order RC Network Transfer Functions*

Modeling nonlinear dynamical systems remains fundamental across many physical domains, where complex interactions often overlay intrinsic linear behaviors. In integrated circuit design, a prominent instance arises in interconnects, where nonlinear drivers interface with passive metal wires forming linear time-invariant (LTI) networks. The passive portion, shaped by parasitic resistances ($R_i$) and capacitances ($C_i$), governs signal delay, attenuation, and waveform distortion. The behavior of these RC networks is typically characterized in the Laplace domain via a transfer function $H(s)$ that maps input signals to output responses [27, 29, 4]. For a network with $m$ independent energy-storage elements, the transfer function takes the form shown in Equation (1):

$$H(s) = \frac{N(s)}{D(s)} = \frac{b_0 + b_1 s + \cdots + b_{m-1} s^{m-1}}{a_0 + a_1 s + \cdots + a_m s^m}.$$
(1)

where the roots of the denominator $D(s)$ are the system poles $p_i = -1/\tau_i = -1/(R_i C_i)$, all located on the negative real axis due to the passive nature of RC circuits.

By the fundamental theorem of algebra and the Heine-Borel theorem, any strictly proper rational function with distinct poles admits a unique partial fraction expansion[30] shown in Equation (2):

$$H(s) = \sum_{i=1}^{m} \frac{r_i}{s - p_i}, \quad r_i = \left. \frac{N(s)}{D'(s)} \right|_{s=p_i}.$$
(2)

This decomposition can be rigorously derived through two classical approaches. First, the method of undetermined coefficients constructs a linear system whose solution is guaranteed by the nonsingularity of the associated Vandermonde matrix when all poles are distinct. Second, the Cauchy residue theorem establishes the residue-based representation by integrating $H(s)$ around a closed contour enclosing all poles; analytic continuation then ensures the uniqueness of this expansion.

Further physical constraints arise from the realizability conditions of RC networks. All poles must lie strictly in the negative real domain to ensure overdamped and stable dynamics. While residues are typically real-valued, they may be either positive or negative, reflecting modal interference effects in higher-order coupled systems. Despite potential non-monotonicity at the modal level, the overall response remains physically consistent and interpretable, capturing the multi-timescale nature of signal propagation inherent to real interconnect behavior.

## 2.2  *Generalization to Repeated Poles*

When a pole $p_i$ of the transfer function has multiplicity $k_i > 1$ (with $\sum_i k_i = m$), the partial-fraction expansion naturally extends to form in Equation (3):

$$H(s) = \sum_{i=1}^{q} \sum_{j=1}^{k_i} \frac{r_{ij}}{(s - p_i)^j}, \tag{3}$$

where each higher-order residue is given by Equation (4):

$$r_{ij} = \frac{1}{(k_i - j)!} \frac{d^{k_i - j}}{ds^{k_i - j}} \left[ (s - p_i)^{k_i} H(s) \right]\Bigg|_{s=p_i}. \tag{4}$$

In practice, however, exact repeated poles are rare in on-chip RC networks due to manufacturing tolerances and layout variations. Even when poles are nearly coincident, the corresponding higher-order residues $r_{ij}$ tend to be small, and their time domain contributions $t^{j-1}e^{p_i t}$ decay rapidly for $p_i < 0$. Consequently, one can safely ignore repeated-pole terms in most modeling tasks and rely on single-pole expansions to achieve high-fidelity simulations [31].

This comprehensive decomposition provides a unified framework for both time-domain and frequency-domain analysis of arbitrary high-order RC interconnect networks.

## 2.3  Computational Complexity Analysis

Traditional SPICE-based transient simulation begins by formulating a system of nonlinear equations based on circuit devices and Kirchhoff's laws. Solving this system—typically via the Newton–Raphson method—is computationally expensive. For an $n$-node RC network, each iteration involves Jacobian construction and LU decomposition, resulting in a complexity of $\mathcal{O}(n^3)$. Even with sparse solvers, fill-in effects lead to an effective cost between $\mathcal{O}(n^{2.5})$ and $\mathcal{O}(n^3)$ over $T$ time steps and $P$ ports [32, 33].

In contrast, our method operates in the S-domain and eliminates matrix inversion. Computing the admittance or impedance to a single output node requires $\mathcal{O}(n)$ operations. Extending this to all $n$ nodes yields a total complexity of $\mathcal{O}(n^2)$, while avoiding iterative linear system solves.

# 3  Data Acquisition and Preprocessing

## 3.1  Simulation Environment Setup

The HSPICE simulation platform employing a 40-nm CMOS PDK ensured process-compliant device parameters ($V_{\text{th}}$, $\lambda$, $I_{\text{leak}}$), where a single-stage CMOS driver with IEEE 1481-2009-compliant RC networks (parasitics extracted via Python) was modeled through state space representation and converted to an $S$-domain transfer function; subsequent partial fraction decomposition yielded first-order subsystems characterized by poles $p_i$ and residues $r_i$ for neural operator-based time domain prediction.

## 3.2  Stimulation Configuration

The input voltage waveform was configured as an ideal step signal (0 to VDD transition). Transient simulations covered both the signal rise phases (0 to 20 ns) and steady state behavior,with a time step resolution of 10ps.

## 3.3  Feature Representation and Supervision

Each sample is defined by a tuple of conditioning inputs and supervised outputs. The conditioning inputs comprise a device type label, a sequence of transient time points $t_1, \ldots, t_T$, and a set of frequency domain features obtained via transfer function decomposition, encoded as pole–residue pairs $\{(p_i, r_i)\}_{i=1}^{m}$. Together, these inputs capture the structural, temporal, and modal characteristics of the circuit.

The supervised target is the voltage response sequence $\{V_{\text{out}}(t_1), \ldots, V_{\text{out}}(t_T)\}$, obtained from HSPICE simulation, which guides training via time-aligned regression.

### 3.4 Feature Normalization

To harmonize heterogeneous input features and enhance model robustness, we apply the following transformations in a single step shown in Equation (5):

$$V'(t) = \frac{V(t) - \min(V)}{\max(V) - \min(V)}, \qquad\qquad t' = \frac{\log_{10}(t) - \mu_t}{\sigma_t} \qquad (5)$$

where $V(t)$ is the original voltage at time $t$, $\min(V)$ and $\max(V)$ are its minimum and maximum over the waveform, $t$ is a sampled transient time point, and $\mu_t, \sigma_t$ are the mean and standard deviation of $\log_{10}(t)$ across all samples. These normalizations place both features on comparable scales, mitigate the influence of outliers, and promote stable, efficient training.

### 3.5 Transfer Function-Based RC Network Modeling

We propose a compact, system-level modeling framework for standard-cell-driven RC interconnects by decomposing the Laplace-domain transfer function shown in Equation (6):

$$H(s) = \frac{V_{\text{out}}(s)}{V_{\text{in}}(s)} == \sum_{i=1}^{n} \frac{A_i}{\frac{s}{p_i} - 1} \qquad (6)$$

where each decay rate $p_i > 0$ (inverse time constant) and residue $A_i \in \mathbb{R}$ satisfies $\sum_i A_i = 1$, ensuring a normalized unit-step response. This form captures the dominant exponential kernels $e^{-p_i t}$ without explicit node-level modeling.

To enable neural-network learning and generalization across circuits of varying size, we encode each mode as a pole-residue pair $(p_i, A_i)$, normalize all $p_i$ and $A_i$ by $\max_i p_i$ and $\max_i |A_i|$, sort pairs by descending $|A_i|$. This interpretable mode sequence accurately reconstructs high-order RC responses with linear complexity and full spectral fidelity.

## 4 Model Architecture Design

### 4.1 Baseline Module: First-Order Prediction

The baseline module predicts the nonlinear voltage response of a single-mode RC system, specified by a pole-residue pair $(p, r)$, at discrete time points $\{t_k\}_{k=1}^{T}$. Unlike ideal linear RC networks, the input waveform here first passes through nonlinear active components (e.g., CMOS drivers), making the overall system response analytically intractable. To address this, we employ a neural function approximator:$\hat{V}(t_k) = f_\theta(p, r, t_k)$.where $f_\theta$ is a lightweight Transformer trained to capture the nonlinear mapping from modal and temporal inputs to voltage outputs[1].

The model consists of three encoder and three decoder layers, each composed of multi-head self-attention, feed-forward sublayers with GELU activation, and layer normalization[34]. Positional encoding is included to preserve temporal structure. Input features $(p, r, t_k)$ are embedded and processed in parallel to predict $\hat{V}(t_k)$ at each time step. The network is trained end-to-end using the AdamW optimizer with weight decay, minimizing the mean squared error[35][36] shown in Equation (7):

$$\mathcal{L}_{\text{MSE}} = \frac{1}{T} \sum_{k=1}^{T} \left( \hat{V}(t_k) - V(t_k) \right)^2 \qquad (7)$$

This architecture provides an accurate and generalizable first-order predictor that forms the foundation for modeling higher-order RC systems through residual correction.

### 4.2 Compensation Module: Residual Correction

The compensation module iteratively refines the baseline prediction by learning the residual error between successive model orders. For each order $n$ and time point $t_k$, the input feature vector is $\left( n, \ p_n, \ r_n, \ p_{n-1}, \ r_{n-1}, \ t_k \right)$.where $(p_n, r_n)$ and $(p_{n-1}, r_{n-1})$ denote the pole-residue pairs for the $n$th and $(n-1)$th modes, respectively.The network outputs shown in Equation (8):

$$\hat{e}_n(t_k) = g_\phi\big(n, p_n, r_n, p_{n-1}, r_{n-1}, t_k\big) \tag{8}$$

which represents the corrective residual to be added to the order-$n$ prediction at time $t_k$.

The function $g_\phi$ is implemented as a lightweight Transformer with the same depth and hyperparameters as the baseline module. It is trained end-to-end using the AdamW optimizer to minimize the residual mean squared error, thereby progressively correcting and refining higher-order predictions.

## 4.3 Recursive Training Procedure

Let $\{(p_i, r_i)\}_{i=1}^n$ be the pole-residue pairs for the $n$th-order model and $\{t_k\}_{k=1}^T$ the sampled time points. Denote by $f_\theta$ the trained first-order predictor and by $\{e_{\phi_j}\}_{j=1}^{n-1}$ the sequence of learned residual modules up to order $n-1$. To train the $n$th residual module $e_{\phi_n}$, we first accumulate the baseline prediction by summing $f_\theta(p_i, r_i, t_k)$ for $i = 1, \ldots, n$. We then add all previously learned corrections $e_{\phi_j}$ to form the current prediction and compute its discrepancy from the true output $V_{\text{out}}(t_k)$. This residual error serves as the target for $e_{\phi_n}$, which is fit by minimizing the mean squared error over $k = 1, \ldots, T$ using the AdamW optimizer. Repeating this process for each order yields a cascade of lightweight modules that progressively refine the high-order RC response.

---

**Algorithm 1** Iterative Residual Correction Training

---

**Require:** Base predictor $f_\theta$, residual module set $\{e_{\phi_j}\}_{j=1}^N$, sampled data $\{t_k, V_{\text{out}}(t_k)\}_{k=1}^T$
**Ensure:** Trained residual modules $\{e_{\phi_j}\}_{j=1}^N$
 1: Initialize cumulative baseline prediction: $\hat{V}_{\text{base}}(t_k) \leftarrow 0, \ \forall k \in [1, T]$
 2: **for** residual index $j = 1$ **to** $N$ **do**
 3:      Load current pole $p_j$ and residue $r_j$
      *Phase 1: Baseline prediction update*
 4:      **for** time step $k = 1$ **to** $T$ **do**
 5:          Update baseline: $\hat{V}_{\text{base}}(t_k) \leftarrow \hat{V}_{\text{base}}(t_k) + f_\theta(p_j, r_j, t_k)$
 6:      **end for**
      *Phase 2: Residual target computation*
 7:      **for** time step $k = 1$ **to** $T$ **do**
 8:          Compute current prediction:

$$\hat{V}_j(t_k) \leftarrow \hat{V}_{\text{base}}(t_k) + \sum_{i=1}^{j-1} e_{\phi_i}(i, p_i, r_i, p_{i-1}, r_{i-1}, t_k)$$

 9:          Compute residual target:

$$r_j(t_k) \leftarrow V_{\text{out}}(t_k) - \hat{V}_j(t_k)$$

10:      **end for**
      *Phase 3: Module training*
11:      Build training set: $\mathcal{D}_j = \{(t_k, r_j(t_k))\}_{k=1}^T$
12:      Minimize the loss:

$$\min_{\phi_j} \frac{1}{T} \sum_{k=1}^T \big(e_{\phi_j}(j, p_j, r_j, p_{j-1}, r_{j-1}, t_k) - r_j(t_k)\big)^2$$

13:      Update $\phi_j$ using gradient descent
14: **end for**

---

By repeating this procedure for $n = 1, 2, \ldots, N$, we ensure each $e_{\phi_n}$ learns to generalize the correction from order $n-1$ to $n$, yielding a cascade of residual models that together approximate the full high-order response with minimal overfitting. Moreover, this recursive training scheme offers a degree of generalization. Each residual module $e_{\phi_n}$ depends only on the pole–residue pairs of two adjacent orders and the current time point, without requiring knowledge of the full network topology

or total number of nodes. As a result, the trained modules can be reasonably extended to refine predictions for moderately higher-order systems beyond those seen during training.

## 4.4 Inference Procedure

In the inference stage, we compute a single forward-pass estimate of the output waveform by first assembling a baseline response and then applying all residual corrections in parallel. Specifically, for each time sample $t_k$ shown in Equation (9):

$$\hat{V}_{\text{base}}(t_k) = \sum_{i=1}^{N} f_\theta(p_i, r_i, t_k) \tag{9}$$

where $p_i$ is the $i$th pole (inverse time constant) of the transfer function, $r_i$ is the corresponding residue (modal weight), and $f_\theta(p_i, r_i, t_k)$ denotes the base predictor's output, typically $r_i e^{-p_i t_k}$.

All $N$ residual modules $e_{\phi_j}$ are then evaluated and summed in parallel shown in Equation (10):

$$\hat{V}_N(t_k) = \hat{V}_{\text{base}}(t_k) + \sum_{j=1}^{N} e_{\phi_j}\left(j, p_j, r_j, p_{j-1}, r_{j-1}, t_k\right) \tag{10}$$

where $e_{\phi_j}(\cdot)$ is the $j$th trained residual correction module, and its inputs $(j, p_j, r_j, p_{j-1}, r_{j-1}, t_k)$ include the current and previous pole-residue pairs as well as the time $t_k$.

This procedure yields the final prediction $\hat{V}_N(t_k)$ at each $t_k$ with $\mathcal{O}(N)$ complexity, running 5–10× faster than commercial tools like HSPICE by requiring just a single pass through the base predictor and residual modules.

# 5 Experiment

## 5.1 Single-Pole Transfer Function: Training and Test Performance

We first evaluate the performance of our model on single-pole transfer functions. The model achieves near-perfect fit on the training data and strong generalization to unseen single-pole functions.

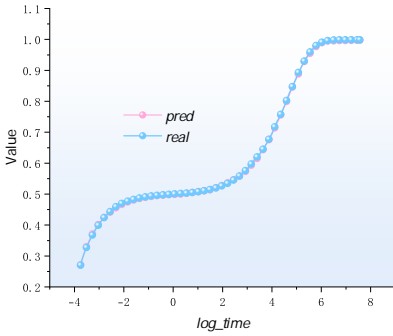

(a) Single-Pole Transfer Function: Training vs. test performance. $R^2 = 0.999$ illustrates near-perfect fit on the training data and strong generalization on held-out single-pole examples.

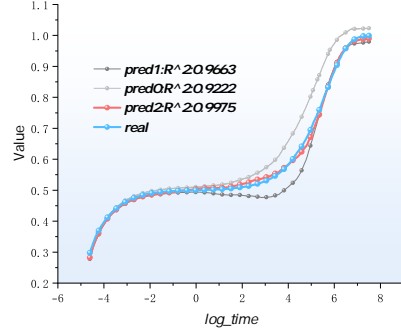

(b) Error-Correction Model: This figure illustrates the step-by-step waveform correction process, from the raw prediction without any error correction through the successive application of first, second, and third-order error modules, culminating in a final fit of $R^2 = 0.9975$

Figure 3: (a) Model performance on single-pole transfer functions, showing minimal overfitting and excellent test-set accuracy. (b) Effectiveness of our recursive error-correction module on a three-pole example.

## 5.2 Effectiveness of the Error-Correction Model

To demonstrate the benefit of our recursive error-correction module, we evaluate on a three-pole transfer function example shown in Equation (11):

$$H(s) \; = \; \sum_{i=1}^{3} \frac{A_i}{\frac{s}{p_i} - 1} \tag{11}$$

where $\{p_i\}$ and $\{A_i\}$ are chosen such that the poles are well separated. Figure 3 shows the predicted and true step responses over time. We compute the coefficient of determination and obtain $R^2 = 0.9975$, confirming that the error-correction stage significantly improves accuracy over the base model.

## 5.3 Generalization to Higher-Order Transfer Functions

Next, we evaluate the model's ability to generalize beyond training orders. We train exclusively on systems of order within 3 and then evaluate on transfer functions of orders from 4 to 9. Table 1 reports MSE and $R^2$ on each higher-order test set. Despite never having seen orders above 3, the model retains strong predictive power, demonstrating effective extrapolation. To probe more extreme extrapolation, we additionally train an error-correction model on 15th-order systems and test it on 200th-order transfer functions; the full setup and per-case results are provided in the appendix. Across 20 held-out cases, the mean $R^2$ reaches 0.983 (see Appendix).

| Order | 4 | 5 | 6 | 7 | 8 | 9 |
|---|---|---|---|---|---|---|
| $R^2$ | 0.986 | 0.997 | 0.995 | 0.990 | 0.979 | 0.964 |

Table 1: Generalization performance on higher-order transfer functions (trained on orders $\leq 3$).

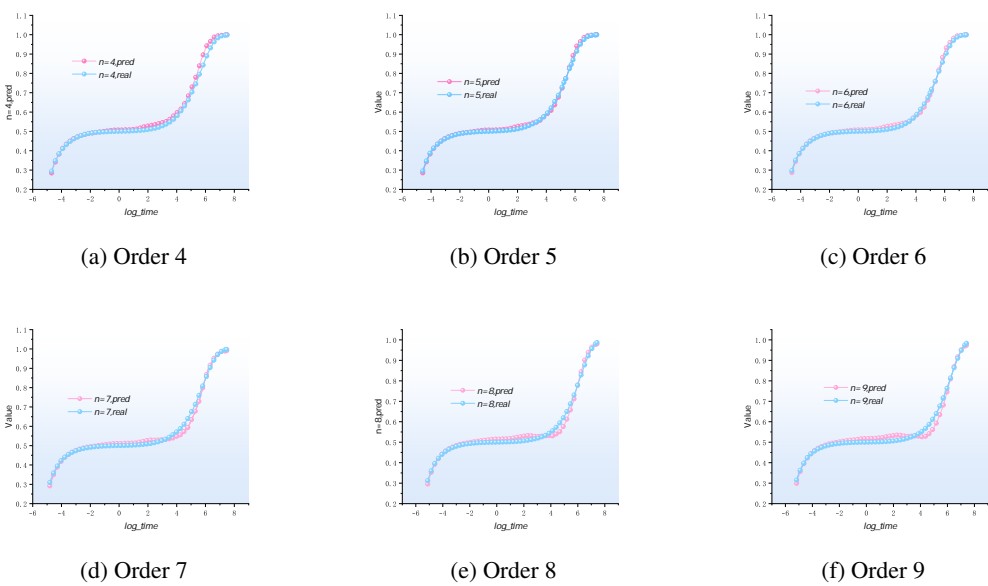

| (a) Order 4 | (b) Order 5 | (c) Order 6 |
|---|---|---|

| (d) Order 7 | (e) Order 8 | (f) Order 9 |
|---|---|---|

Figure 4: Prediction vs. true response for transfer-function orders 4 to 9.

## 5.4 Inference Time Comparison

**Training cost.** Each residual-correction module contains approximately 100K–160K parameters. All models were trained on a single NVIDIA RTX 4090 GPU. For reference, single-pole systems typically train in under 20 minutes, while 10th-order systems converge in approximately 2 hours. Given the considerable acceleration our model achieves at inference time, this one-time training cost is highly acceptable.

**Dataset construction.** Training data is generated using standard *HSPICE* simulations. Each simulation typically completes in about 0.5 seconds. For our full training dataset of $\sim 5{,}000$ samples, the

end-to-end preprocessing pipeline (simulation, waveform extraction, and supervised data formatting) completes in under 2 hours. The process is fully scriptable and parallelizable, and does not pose a practical bottleneck.

**Runtime comparison.** Finally, we compare the runtime of our S-Crescendo model against *HSPICE* on a 10 ns transient simulation sampled at 1,000 time steps. *HSPICE* runtimes for orders 1 through 10 were measured on a CPU node equipped with an AMD EPYC 7763 (64 cores) and 256 GB of DDR4 RAM; S-Crescendo inference times were recorded on a workstation with an NVIDIA RTX 4090 GPU. Table 2 reports the average simulation and inference times. Across all orders, S-Crescendo achieves more than two orders of magnitude speedup while preserving high accuracy.

**DCM baseline.** Dynamic Circuit Macromodeling (DCM) is a physics-inspired model-order reduction technique that constructs dynamic macromodels to balance accuracy and efficiency. On the same 10-order RC test file, DCM completes in 0.6 s with a fit accuracy of $R^2 = 0.9983$.[1]

**NGSPICE baseline.** *NGSPICE* is a widely used open-source SPICE simulator adopted across academia and industry. Although typically slower than commercial tools such as *HSPICE*, its accessibility and device-level accuracy make it a credible practical reference. On our 10-order RC test file, *NGSPICE* completes in 1.08 s and—because it shares core numerical algorithms with *HSPICE*—delivers equivalently high fidelity (we did not compute a separate $R^2$ against *HSPICE*).[2]

*Note.* Order-10 is the *slowest* inference case for our model; benchmarking against this conservative worst case still yields the above margins over DCM, NGSPICE, and HSPICE, underscoring the strength of our approach.

Table 2: Runtime comparison between HSPICE and S-Crescendo across transfer function orders (10ns, 1000 steps).

| Order | 1 | 2 | 3 | 4 | 5 | 6 | 7 | 8 | 9 | 10 |
|---|---|---|---|---|---|---|---|---|---|---|
| HSPICE (s) | 0.26 | 0.21 | 0.23 | 0.22 | 0.27 | 0.26 | 0.28 | 0.23 | 0.23 | 0.26 |
| S-Crescendo (s) | 0.014 | 0.019 | 0.022 | 0.018 | 0.023 | 0.028 | 0.031 | 0.034 | 0.039 | 0.042 |
| Speedup (X) | 18.6 | 11.1 | 10.5 | 12.2 | 11.7 | 9.3 | 9.0 | 6.8 | 5.9 | 6.2 |

# 6 Further Discussion

## 6.1 Degradation of Accuracy at Higher Orders

While S-Crescendo performs well on low to mid-order transfer functions, its $R^2$ degrades as order $m$ increases. This is due to recursive error accumulation: each correction module $\varepsilon_k(t)$ adjusts not only current residuals but also propagates previous errors. If the single-pole state space has size $N$, then the full $m$-pole space grows as $N^m$, whereas each error model sees only $N$ samples during training. Thus, the fraction of covered states is shown in Equation (12):

$$\frac{N \times m}{N^m} \tag{12}$$

which shrinks rapidly with $m$, leading to sparse supervision and compounded inaccuracies.

## 6.2 Reducing Data Dependency via Blocked Recursion

To alleviate error accumulation and data explosion, we propose a blocked recursion strategy. Instead of training a separate module per order, we group adjacent orders into blocks—for example, a shared module for orders 2–4, another for 5–6, etc. Each block is supervised on $O(N \times B)$ states (for block size $B$), yet extrapolates across $B$ orders. This reduces the number of recursive calls and lowers training demands, improving both runtime and scalability.

---

[1]All DCM runtimes were measured on a machine with AMD EPYC 7763 (64 cores), 1 TB RAM, and 30 GB swap.

[2]NGSPICE runtime was measured on a machine with Intel Core i7-14650HX CPU and 32 GB RAM. S-Crescendo inference uses the same RTX 4090 workstation described in this paper.

### 6.3 Scaling of Inference Latency with Order

S-Crescendo's inference latency scales roughly linearly with order $m$, since each pole adds a forward pass. In contrast, tools like HSPICE collapse high-order dynamics via model reduction, maintaining near-constant runtime. To bridge this gap, we consider (i) collapsing low-impact poles into aggregate corrections, or (ii) integrating model-order reduction into the learned pipeline—both strategies aim to extend our efficiency gains to large-scale systems.

### 6.4 Modeling Limitation: Repeated Poles in Transfer Functions

The model currently does not handle repeated poles, which introduce higher-order temporal terms like $t^k e^{\lambda t}$. To address this, future models can extend input features to include pole multiplicity, enabling learning from triplets $(p_i, A_i, m_i)$. This would broaden the model's applicability to more complex, higher-order dynamics.

### 6.5 Outlook: Toward General Nonlinear–Linear Hybrid Systems

Beyond RC modeling, the proposed framework extends naturally to hybrid systems with a "nonlinear front-end + linear dynamic core" architecture, common across engineering domains. Examples include switching converters in power electronics, where nonlinear control drives linear filters; analog front-ends, where transistor drivers interface with RC loads; and neural membrane models coupling nonlinear ion channels to capacitive elements. By modeling linear dynamics via Laplace-domain priors and learning nonlinear corrections, the framework enables efficient, interpretable emulation. Future directions include integrating operator learning or domain-specific constraints to broaden applicability.

## 7 Acknowledgments and Disclosure of Funding

This work was supported by the National Natural Science Foundation of China (NSFC) under Grant 62374189 and Guangdong Innovation and Entrepreneurship Team Project under Grant Innovation Strategy (2021ZT09X070).

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

# Appendix Contents

# A  Related Works

## A.1  Background

Modeling and simulation of high-order nonlinear systems remain central challenges in modern VLSI backend design, particularly in analog/mixed-signal verification and large-scale circuit time-domain modeling. Mainstream industrial tools such as HSPICE predominantly rely on Newton-Raphson iterative solvers coupled with numerical integration schemes like Gear or Trapezoidal methods. While these solvers offer high physical accuracy, their simulation time and resource consumption scale poorly with system complexity-especially in the presence of strong nonlinearities or high-order dynamics-posing significant computational bottlenecks for practical deployment.

In recent years, neural networks have demonstrated remarkable success across a wide range of scientific computing applications, including areas within electronic design automation (EDA) such as device modeling [37], analog circuit fault detection [38], and high-frequency electromagnetic simulation [39]. However, for the specific task of modeling signal line RC responses in high-order nonlinear systems, effective deep learning-based approaches remain largely unexplored. To the best of our knowledge, no existing work has leveraged deep neural networks-particularly Transformer-based architectures-for direct time-domain modeling of such systems.

Given this gap, we provide a comprehensive review of related work from two perspectives: (1) traditional modeling approaches developed in the EDA community for approximating RC signal line responses, including current source models, voltage response models, and direct waveform fitting methods; and (2) recent advances in neural network-based methods for modeling general nonlinear dynamical systems, which-while related in scope-target different modeling granularities and lack direct applicability to high-order S-domain waveform prediction.

## A.2  Traditional Methods for Signal Line RC Response Modeling

Existing methods for modeling signal line RC response can be broadly classified into three categories:

The first category is the Current Source Model (CSM). Criox and Wong proposed a gate cell current source model called Blade [20], which consists of a voltage-controlled current source, internal capacitance, and a one-step time-shift operation. Kellor further enhanced model accuracy by introducing the KTV model [40], which considers Miller capacitance. Subsequently, Li and Acar [41] and Fatemi et al. [42] introduced input and output parasitic capacitances, modeling the output current source as a function of input/output voltages, gradually incorporating nonlinear characteristics into CSM models. However, since CSM-based methods can only match fixed effective capacitances (up to two) throughout the process, the simulation accuracy of current/voltage waveforms is inherently limited [24][43][44][45][46][47] . In recent years, widely adopted industry methods such as Composite Current Source (CCS) [21] and Extended Current Source Model (ECSM) [48] have established driver and receiver models for each cell to handle scenarios with nonlinear input and crosstalk. Nevertheless, CSM-based approaches still face significant challenges in matching high-order RC load characteristics, limiting their accuracy in current response prediction.

The second category is the Voltage Response Model (VRM), such as the Non-Linear Delay Model (NLDM). Iterative methods [22] [25] [49] [50] , although capable of achieving high precision, often require substantial CPU time for convergence. Non-iterative methods [51] [52], on the other hand, rely on closed-form expressions that offer faster computation but can result in output waveform matching errors of up to 15% [26]. Furthermore, as technology nodes shrink and RC loads become more complex, two-parameter fitting methods struggle to accurately capture the response curve of RC networks, limiting their applicability in high-precision simulations [53].

The third category consists of Direct Waveform Prediction Methods, such as double exponential functions [23], Weibull functions [54], and gamma functions [55], which directly fit the current or voltage response. Recently, a macromodeling method [56] was proposed that uses SPICE to extract parameters for modeling, which improves accuracy to some extent. However, these direct fitting methods are unable to predict initial overshoot/undershoot effects, which become more pronounced when the input slope is large.

### A.3  Current Applications of Neural Networks in Modeling Nonlinear Systems

In recent years, neural networks have been widely applied to modeling and prediction tasks of nonlinear dynamical systems. Neural ODE methods [12] introduce continuous-time differential equations to model system evolution, which are suitable for trajectory prediction problems. Physics-Informed Neural Networks (PINNs) [11] incorporate partial differential equation constraints during training to enhance physical consistency of the model. Additionally, Reservoir Computing [57] has demonstrated excellent performance in modeling short-term behavior of chaotic systems, while Koopman operator learning [58, 59] attempts to linearize nonlinear systems by mapping them into a linear space to simplify modeling. Graph Neural Networks (GNNs) [60] process complex local interactions by constructing graphs among components, and neural operator methods, such as the Fourier Neural Operator (FNO) [14] and DeepONet [13], focus on learning mappings of high-dimensional functions. In the domain of time series forecasting, Transformer-based models and their variants (e.g., Informer [61]) have demonstrated strong capability in modeling long-term dependencies. Although these methods have achieved substantial progress in various scientific computing scenarios, existing works mainly focus on low-dimensional systems, continuous trajectory modeling, or frequency-domain PDE solving, with limited exploration in modeling high-order time-domain responses of complex circuit RC networks.

The proposed S-Crescendo framework in this work possesses three key characteristics designed to effectively model time-domain behaviors in high-order nonlinear systems. First, it explicitly separates system modes via partial fraction decomposition of the S-domain transfer function. Second, it employs a distributed Transformer architecture combined with pole-residue embedding strategies, reducing the response prediction complexity from cubic $\mathcal{O}(n^3)$ to linear $\mathcal{O}(n)$. Third, it introduces an attention-based correction operator that adaptively captures nonlinear coupled responses while modeling dominant modes. This framework combines physical interpretability with computational efficiency and demonstrates high-fidelity waveform fitting consistent with HSPICE simulations on validation datasets, thus filling the gap of deep learning-based modeling for high-order circuit systems.

## B  Dataset Preparation Details

### B.1  Reference Simulator for Ground-Truth Generation

All training and evaluation datasets in this work are generated using Synopsys PrimeSim HSPICE® U-2023.03-SP2-2. HSPICE is widely acknowledged as the de facto industry reference for analog and mixed-signal circuit simulation, owing to its consistently strong agreement with post-silicon measurements—typically achieving within 1%-5% error across a broad range of process nodes (from 0.18 µm to 3 nm). It provides foundry-certified transistor and passive device models, enabling high physical fidelity and process portability. Moreover, its Precision Parallel simulation engine supports near-linear scalability on multi-core systems such as the AMD EPYC 7763 (64-core), facilitating efficient, high-accuracy waveform generation even for large-scale nonlinear networks. These capabilities establish HSPICE as a trusted ground-truth generator for validating data-driven modeling frameworks.

### B.2  Dataset Computing Environment

The experiments were conducted on a computing platform running CentOS Linux 7, an operating system known for its long-term stability and high compatibility with the source code of Red Hat Enterprise Linux (RHEL). This ensures the reliability and consistency of the system environment, facilitating reproducibility of the experimental results.

The hardware platform is equipped with an AMD EPYC 7763 processor, featuring 64 physical cores with a base clock frequency of 2.45 GHz and dynamic boost up to 3.5 GHz. It includes a 256 MB L3 cache and an 8-channel DDR4-3200 MT/s memory architecture, providing substantial parallel computing capability and high memory bandwidth. This configuration offers ample computational resources and efficient data handling, guaranteeing smooth execution of large-scale simulation workloads while minimizing potential performance bottlenecks.

## B.3    Dataset Preparation

The dataset was constructed through a multi-stage pipeline. First, a Python script automatically generated SPICE netlists (".sp" files) representing RC networks of varying orders. These netlists were then processed along two parallel paths. In the first path, a Python module parsed each netlist to compute the corresponding analytical transfer function. In the second path, all netlists were batch-simulated using PrimeSim HSPICE U-2023.03-SP2-2 to obtain time-domain simulation results. Upon completion of the simulations, another Python script extracted the relevant outputs from the HSPICE raw data files. Finally, the analytical transfer functions and the corresponding simulated results were aligned and merged into a unified dataset, which served as the basis for subsequent model training and evaluation (see Figure 5).

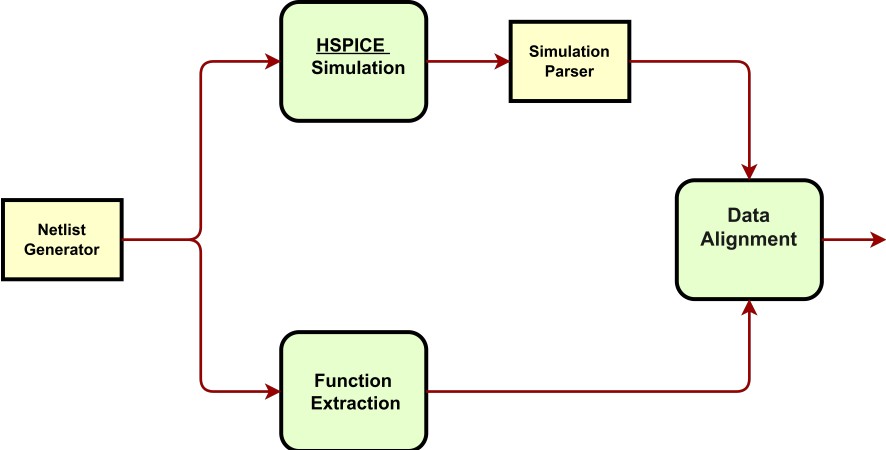

Figure 5: Overview of the dataset preparation pipeline.

### B.3.1    Netlist Generation

A Python script automatically generates SPICE netlists for cascaded RC networks of arbitrary order. For first-order RC circuits, we adopt an exhaustive grid-based method by varying $R_1$ and $C_1$ across predefined intervals to construct a comprehensive dataset that covers a wide range of transfer functions. The script iterates over discrete values and generates files named `testsuite_<n>.sp`, each invoking a transient simulation (`.TRAN 10ps 20ns`) and including standard PDK models to ensure consistency.

For higher-order RC networks ($N > 1$), due to the exponential growth in parameter space, exhaustive enumeration becomes computationally infeasible. Instead, we construct a continuous state space defined by all possible tuples $\{R_n, C_n\}_{n=1}^N$, and sample from this space using randomized or quasi-random strategies. Each sample corresponds to a specific configuration of RC parameters, which is then used to generate a netlist representing an $N$th-order cascaded RC system. This hybrid strategy ensures both completeness in low-order cases and scalability in high-order scenarios, providing a diverse and representative dataset for model training and evaluation.

The dataset generation scripts and usage examples are available at our code in supplemental_material.zip.

**Circuit structure**    We construct a canonical nonlinear–linear cascade by driving a linear RC network with a nonlinear inverter. The nonlinear frontend is implemented as a standard-cell inverter in SPICE:

```
X1 I1 OUT vdd 0 CLKINV3_12TR50
```

Here, `OUT` is the inverter's output node, which serves as the input to the linear RC load. In the simplest first-order configuration, the RC network consists of a single resistor–capacitor pair:

```
R1 OUT        NODE1   r1
C1 NODE1      0       c1
```

To generalize this to an $N$-stage cascade, we chain $N$ identical first-order sections. Each section $n$ is defined in the netlist as:

```
R1 OUT       NODE1    r1
C1 NODE1     0        c1

⋮

R<n> NODE<n-1> NODE<n> r<n>
C<n> NODE<n>   0        c<n>
```

where the nodes are labeled sequentially as `NODE0=OUT`, `NODE1`, ..., `NODEn`, and `NODEn` is grounded.

Although a single stage has the well-known transfer function

$$H_n(s) = \frac{1}{1 + sR_nC_n},$$

the overall transfer function of the cascaded network cannot be written as the simple product $\prod_{n=1}^{N} H_n(s)$. Inter-stage coupling—where each stage's output drives the next—requires solving the full circuit equations or deriving a state-space model to obtain $H(s)$ correctly.

Once the high-order transfer function $H(s)$ is obtained, we apply *partial fraction expansion* (PFE) to decompose it into modal contributions:

$$H(s) = \sum_{k=1}^{N} \frac{r_k}{s - p_k},$$

where $p_k$ and $r_k$ denote the poles and residues, respectively. This modal form explicitly exposes the dynamic modes of the network and forms the theoretical foundation for our S-domain–aware Transformer model in time-domain prediction of high-order nonlinear circuits.

### B.3.2 Automated HSPICE Simulation and Simulation Parser

The HSPICE simulation workflow was fully automated via a Python script that sequentially runs all SPICE netlists. Each simulation outputs transient voltage responses at the designated output node, saved in `.lis` files containing detailed time-domain voltage and current waveforms.

To facilitate efficient data extraction and downstream processing, a dedicated Python parser was implemented to systematically extract voltage response sequences from these `.lis` files. A representative snippet of the extracted data is shown below:

```
time          voltage
0.000000e+00  1.8579u
1.000000e-11  9.7816m
2.000000e-11  1.78649e-02
3.000000e-11  2.54157e-02
4.000000e-11  3.26519e-02
5.000000e-11  3.79554e-02
6.000000e-11  4.32588e-02
```

These extracted voltage traces serve as ground truth data for validating the analytical transfer functions and for subsequent model training and evaluation.

### B.3.3 Function Extraction

To characterize the linear dynamics of cascaded RC networks, we implemented a fully automated Python pipeline comprising four stages: netlist parsing, state-space construction, transfer-function computation, and result serialization.

**1. Netlist Parsing** Using regular expressions, the parser reads each SPICE file to extract resistor $R_i$ and capacitor $C_i$ values. For a third-order network, the script locates lines beginning with `R1`, `R2`, ..., `C3` and applies unit-aware conversion (e.g., $f \rightarrow 10^{-15}$, $u \rightarrow 10^{-6}$). Missing or malformed entries trigger an exception to ensure data integrity.

**2. State-Space Model Construction**   Given the extracted $\{R_i, C_i\}$, an admittance matrix $G \in \mathbb{R}^{3 \times 3}$ and capacitance matrix $C_{\text{diag}}$ are assembled for the 3-stage RC ladder. The continuous-time state-space matrices are then computed as

$$\mathbf{A} = -C_{\text{diag}}^{-1} G, \quad \mathbf{B} = C_{\text{diag}}^{-1} [\, 1/R_1, \, 0, \, 0 \,]^{\mathsf{T}}, \quad \mathbf{C} = [\, 0, \, 0, \, 1 \,].$$

**3. Transfer-Function Computation**   The transfer function $H(s)$ is obtained from $(\mathbf{A}, \mathbf{B}, \mathbf{C})$ via SciPy's `ss2tf` routine, yielding numerator and denominator polynomials. We apply partial-fraction expansion (`residue`) to extract poles $p_i$ and residues $r_i$. After filtering negligible imaginary parts, poles are sorted by magnitude, and normalized coefficients are computed as

$$A_i' = -\frac{r_i}{p_i}, \quad A_i' \leftarrow \frac{A_i'}{\sum_j A_j'}.$$

**4. Result Serialization**   For each netlist, the tuple $(A_1', p_1, A_2', p_2, A_3', p_3)$ is written as a single row in a CSV file. This standardized format enables downstream training pipelines to ingest model parameters directly.

## B.4   Data Alignment

To prepare the dataset for sequence modeling tasks, we concatenate the static circuit features with the time-dependent voltage response while preserving the temporal dimension explicitly.

Let $\mathbf{x} \in \mathbb{R}^{2n}$ denote the RC feature vector extracted from an $n$-th order RC network (e.g., normalized residues and poles). For a third-order network, we have:

$$\mathbf{x} = (A_1', p_1, A_2', p_2, A_3', p_3) \in \mathbb{R}^6.$$

Let the voltage response sequence over $T$ time steps be represented as a set of timestamped scalar pairs:

$$\{(t_1, y_1), (t_2, y_2), \ldots, (t_T, y_T)\}, \quad \text{where } y_t \in \mathbb{R}, \; t_t \in \mathbb{R}.$$

To incorporate both dynamic and static information, we replicate the static vector $\mathbf{x}$ at each time step and form the augmented matrix:

$$\mathbf{z}(t) = \begin{bmatrix} t_1 & y_1 & \mathbf{x} \\ t_2 & y_2 & \mathbf{x} \\ \vdots & \vdots & \vdots \\ t_T & y_T & \mathbf{x} \end{bmatrix} \in \mathbb{R}^{T \times (2+2n)}.$$

This results in a fully timestamped sequence $\mathbf{z}(t)$, where each row consists of:

- the current simulation time $t_t$,
- the corresponding voltage response $y_t$,
- and the circuit's physical parameters $\mathbf{x}$.

Such a format allows temporal models to condition predictions not only on voltage dynamics but also on circuit-specific properties.

All datasets were automatically constructed using a Python script that fuses the time vector, voltage response, and RC features.

# C   Log-Centered Time Warping & Uniform Resampling

## C.1   Motivation

A step response $v(t)$ typically exhibits an extremely steep leading edge followed by a long, almost flat settling tail. When the raw waveform is sampled on a *linear* time axis, the number of informative points located on the rising edge is orders of magnitude smaller than the points on the tail. Consequently, regression models trained with a uniform loss (e.g. MSE) tend to *under-fit* the neighbourhood of the edge and over-fit the low-slope region, yielding poor predictions for signals whose rise times differ markedly inside the same simulation window.

## C.2 Center detection

Let $t_c$ be the time at which the normalised voltage first crosses $v = 0.5$:

$$v(t_c) = \tfrac{1}{2}V_{\text{DD}}, \qquad t_c = \arg\min_t \left| v(t) - \tfrac{1}{2}V_{\text{DD}} \right|.$$

The algorithm finds $t_c$ with a two–stage linear B-spline interpolation around the crossing (see `find_t()` in the accompanying code base).

## C.3 Log-centred warping

For every sample $t$ we define a *log-centred* time coordinate

$$\tau = \text{sgn}(t - t_c) \left[ \ln\left( |t - t_c|/T_0 + \varepsilon \right) - \ln\varepsilon \right], \tag{13}$$

with scale factor $T_0 = 10^{-10}$ s (empirically chosen) and numerical guard $\varepsilon = 0.1$. Equation (13) is *odd* around $t_c$ and strictly monotone; consequently the mapping $t \mapsto \tau$ is invertible and preserves temporal order.

**Slope compression.** The first derivative

$$\frac{d\tau}{dt} = \frac{1}{|t - t_c| + \varepsilon T_0}$$

is large when $|t - t_c|$ is small and diminishes as $|t - t_c| \to \infty$. Hence points on the steep edge ($|t - t_c| \ll T_0$) are *stretched* in $\tau$-space while the quasi-flat tail is *compressed*. In effect, the dynamic range of local slopes

$$\left| \tfrac{dv}{dt} \right| \longrightarrow \left| \tfrac{dv}{d\tau} \right| = \left| \tfrac{dv}{dt} \right| \frac{dt}{d\tau} = \left| \tfrac{dv}{dt} \right| \left( |t - t_c| + \varepsilon T_0 \right)$$

is equalised, yielding a better-conditioned learning target.

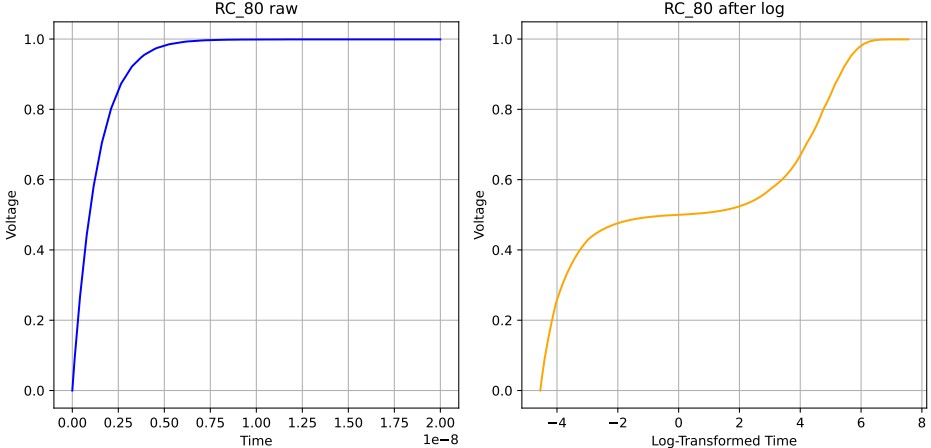

Figure 6: Raw waveform vs. log-centred waveform.

## C.4 Uniform resampling in the warped domain

After warping, we resample the trace at $N$ *equally spaced* $\tau$-locations

$$\tau_k = \tau_{\min} + k\Delta\tau, \quad k = 0, \dots, N-1, \qquad \Delta\tau = \frac{\tau_{\max} - \tau_{\min}}{N-1}.$$

Linear interpolation in $\tau$-space is equivalent to *non-uniform* interpolation in the original time domain, so the final dataset allocates identical representational capacity to equal increments of $\tau$, i.e. to equal *log-scaled* time gaps with respect to the edge.

Figure 7 shows: (a) The original voltage response $v(t)$ for the first RC channel plotted against real time $t$. (b) The same trace after applying the center-log warp $t \mapsto \tau$ and uniformly resampling in $\tau$-space. Black circles indicate the new sample locations, illustrating the denser coverage near the rising edge and sparser coverage on the tail.

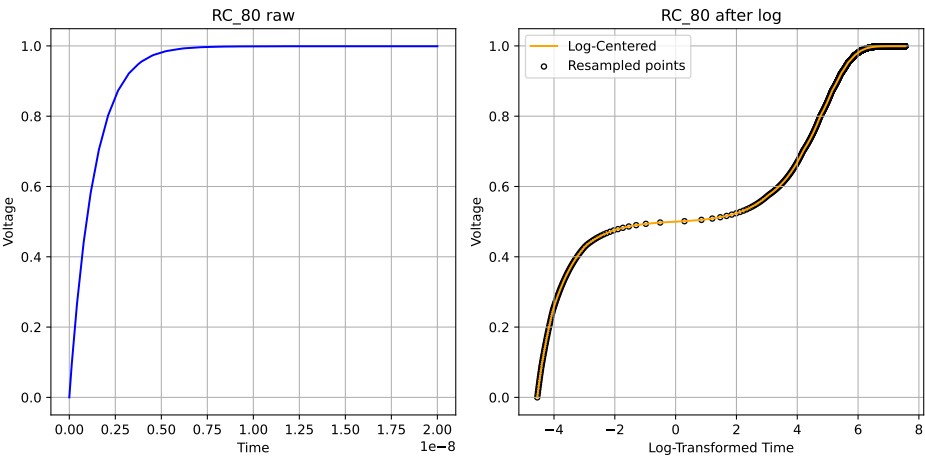

Figure 7: Scatter plot in $\tau$-space: before vs. after resampling.

Comparison of the original step response and the uniform resampling in log-time domain. The blue markers show the raw voltage trace $v(t)$ for the RC network sample (from Figure 8), plotted against the log-centered time coordinate $\tau$. The orange markers overlay the uniformly spaced samples in $\tau$-space, demonstrating that the resampling allocates more points near the steep rising edge while compressing the long tail.

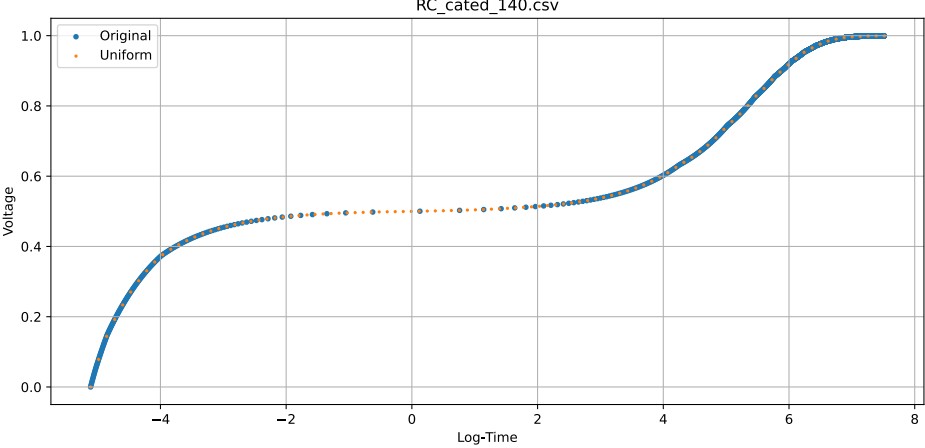

Figure 8: Scatter plot in $\tau$-space: resampling vs. after uniformed resampling.

## C.5 Effect of Uniform Resampling on Regression Accuracy

We conducted an experiment to quantify the impact of our log-centered warping *plus* uniform resampling on step-response modelling accuracy. Under identical training conditions: 60 epochs, batch size = 1000, using 80 training files (80 × 1000 samples) drawn from the full set of 100 RC cases. The model was evaluated on the held-out test set comprising cases RC_81 through RC_100. We report the coefficient of determination $R^2$ for predictions made with ("Resampled") and without ("Raw") the uniform resampling in warped time.

Table 3: $R^2$ comparison: raw vs. log-centered $\tau$-uniform resampling.

| Case | $R^2$ Raw | $R^2$ Resampled |
|------|-----------|-----------------|
| RC_81 | 0.9571 | 0.9998 |
| RC_82 | 0.9591 | 0.9998 |
| RC_83 | 0.9609 | 0.9998 |
| RC_84 | 0.9629 | 0.9998 |
| RC_85 | 0.9647 | 0.9998 |
| RC_86 | 0.9662 | 0.9998 |
| RC_87 | 0.9675 | 0.9998 |
| RC_88 | 0.9690 | 0.9998 |
| RC_89 | 0.9702 | 0.9998 |
| RC_90 | 0.9714 | 0.9998 |
| RC_91 | 0.9725 | 0.9998 |
| RC_92 | 0.9736 | 0.9998 |
| RC_93 | 0.9745 | 0.9998 |
| RC_94 | 0.9753 | 0.9998 |
| RC_95 | 0.9761 | 0.9998 |
| RC_96 | 0.9768 | 0.9998 |
| RC_97 | 0.9774 | 0.9998 |
| RC_98 | 0.9780 | 0.9998 |
| RC_99 | 0.9785 | 0.9999 |
| RC_100 | 0.9816 | 0.9999 |
| **Mean** | 0.9715 | 0.9998 |

Qualitatively, Table 3 shows that uniform resampling in $\tau$-space consistently boosts $R^2$ from the high-0.95 range up to nearly perfect 0.999. On average, the preprocessing yields an absolute improvement of over 0.028 in $R^2$, demonstrating that our log-centered warp and uniform sampling dramatically enhances the model's ability to fit diverse step-response curves under identical training regimes.

As shown in Figure 9, we compare the predictive accuracy of our network on the same RC trace under two preprocessing regimes. In subfigure 9a, the model is trained and evaluated directly on the raw time-domain samples, achieving an $R^2$ of 0.9571. While the overall step response shape is captured, the prediction exhibits noticeable lag on the steep rising edge and slight deviation in the mid-tail region. In contrast, subfigure 9b illustrates the result after applying the center-log warping followed by uniform resampling in the warped $\tau$-domain. Here, the fit improves dramatically to an $R^2$ of 0.9998, with the predicted curve (orange) virtually indistinguishable from the ground truth (blue) across both the fast edge and the extended settling tail. This comparison clearly demonstrates that our log-centering and uniform resampling pipeline substantially enhances the model's ability to learn and generalize step-response dynamics under identical training conditions.

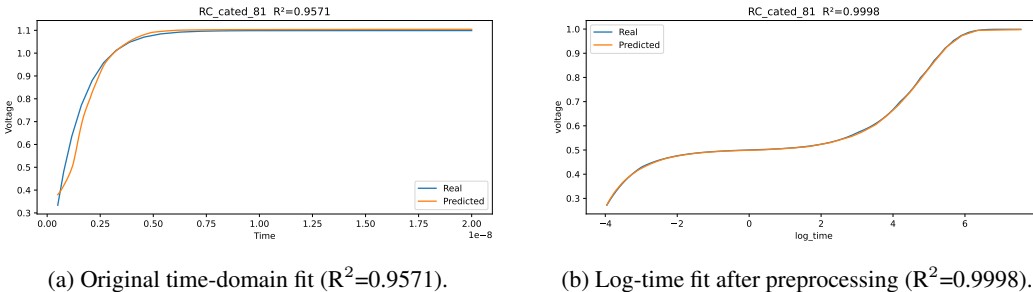

(a) Original time-domain fit ($R^2$=0.9571).    (b) Log-time fit after preprocessing ($R^2$=0.9998).

Figure 9: Model predictions vs. ground truth for two RC traces: (a) raw time-domain input, (b) preprocessed (center-log warp + uniform resampling) input.

## C.6 Theoretical impact on learning

- **Variance reduction.** Let $\sigma^2_{\text{raw}}$ denote the variance of the target derivative $\frac{dv}{dt}$ on the raw grid and $\sigma^2_{\text{warp}}$ the variance of $\frac{dv}{d\tau}$ on the warped grid. From the slope-compression term above it follows that $\sigma^2_{\text{warp}} \leq \sigma^2_{\text{raw}}$, reducing heteroscedasticity seen by the learner.

- **Effective sample size (ESS).** The warping acts as an *importance-sampling* scheme with weight $w(t) = |t - t_c| + \varepsilon T_0$. ESS increases because highly informative edge samples receive larger weights after transformation.

- **Improved Lipschitz constant.** Denote the network $f_\theta$ with Lipschitz constant $L$. The composite function $f_\theta \circ g^{-1}$ (where $g^{-1}$ is the inverse warp) retains the same $L$ but is now evaluated on a domain where the target variation is smaller, tightening generalisation bounds.

## C.7 Proof of Lipschitz Constant Reduction

We now show that, under mild parameter choices, the pre-/post-warp composite

$$F(\tau) \;=\; f_\theta\big(g^{-1}(\tau)\big)$$

has an overall Lipschitz constant $L' < L$.

**1. Composition lemma.** If $f : X \to Y$ is $L_f$-Lipschitz and $h : Z \to X$ is $L_h$-Lipschitz, then

$$\|f \circ h(z_1) - f \circ h(z_2)\| \;\leq\; L_f \, \|h(z_1) - h(z_2)\| \;\leq\; L_f \, L_h \, \|z_1 - z_2\|,$$

so $f \circ h$ is $L_f L_h$-Lipschitz [62].

**2. Derivative of the inverse warp.** Recall the forward warp

$$\tau = g(t) = \text{sgn}(t - t_c)\Big[\ln\big(\tfrac{|t-t_c|}{T_0} + \varepsilon\big) - \ln\varepsilon\Big].$$

Differentiating,

$$\frac{d\tau}{dt} = \frac{1}{|t - t_c| + \varepsilon T_0} \;\implies\; \frac{dt}{d\tau} = |t - t_c| + \varepsilon T_0.$$

Since a one-dimensional $C^1$ function $h$ with $\sup |h'(x)| \leq M$ is $M$-Lipschitz by the Mean Value Theorem [63], it follows that

$$L_{g^{-1}} = \sup_{\tau \in [\tau_{\min}, \tau_{\max}]} \left|\frac{d\,g^{-1}}{d\tau}\right| = \sup_{t \in [t_{\min}, t_{\max}]} \big(|t - t_c| + \varepsilon T_0\big).$$

**3. Parameter choice for contraction.** Let $\Delta t = \max\{|t_{\max} - t_c|, |t_{\min} - t_c|\}$. If we choose parameters so that

$$\Delta t + \varepsilon T_0 < 1,$$

then

$$L_{g^{-1}} < 1 \quad \implies \quad L' = L_{f_\theta} L_{g^{-1}} < L_{f_\theta}.$$

Hence $F(\tau)$ is strictly more contractive than $f_\theta$ on the original $t$-domain.

**4. Impact on generalisation.** Standard Rademacher-complexity generalisation bounds scale linearly with the Lipschitz constant of the hypothesis class and inversely with $\sqrt{N}$ [64, 65]. By reducing the effective Lipschitz constant from $L$ to $L' < L$, we tighten the bound

$$\mathcal{E}_{\text{gen}} = O\Big(\tfrac{L'}{\sqrt{N}}\Big) \;\subset\; O\Big(\tfrac{L}{\sqrt{N}}\Big),$$

thereby improving expected test performance.

## C.8 Implementation details

1. **Voltage normalisation.** All voltages are scaled by $V_{\text{DD}}$ so that $v \in [0, 1]$.

2. **Time origin.** Warping is performed *after* shifting the origin to $t_c$. This removes sample-to-sample phase variation.

3. **Parameter choices.** $T_0$ controls the width of the expanded region; in our experiments $(10^{-10}s)$ adequately covers modern technology nodes down to 3 ps edges.

# D   Full Test Case

## D.1   Training Data Preparation

### Netlist Generation

- Prepare a pool of base RC parameter pairs

- Run:

```
python prepare/n.py \
    --order n \
    --count m \
    --output_dir sp_files_n
```

### HSPICE Simulation

- Simulations are executed via `sim.sh` under Linux.

- Run:

```
hspice testsuite_<n>.sp -o lis/result_<n>
```

Simulation outputs are stored in the `lis/` directory.

### Simulation Parser

- Place your HSPICE `.lis` files into the `./result/` directory.

- Run the parsing script to extract time-voltage waveforms and convert units:

```
python parse_spice.py \
--input_dir ./result \
--output_dir ./result/csv/
```

- Parsed CSV files will be saved in `./result/csv/` for further analysis.

### Function Extraction

- Place your SPICE netlist files (`.sp`) into the directory `sp_files_n/`.

- Run the extraction script to parse resistor and capacitor values, compute state-space parameters, poles, and residues:

```
python extract_functions.py \
--input_dir sp_files_n \
--output_dir result_n
```

- The extracted normalized residues $A_i'$ and poles $p_i$ for each netlist are saved as CSV files in `result_n/`.

### Data Alignment

- Place your RC parameter CSVs in `RC/` and parsed SPICE result CSVs in `lis/result/`.

- Run the script to merge each pair into `result/` as follows:

```
python data_alignment.py
```

- This script reads the first row from each RC CSV, duplicates it to match the length of the corresponding SPICE CSV, concatenates them column-wise, and saves the combined CSV.

- Processed files are named `RC_cated_i.csv` for $i = 1, \ldots, n$.

## D.2 First-Order Model Training and Error-Model Generation

The following workflow describes how to train the first-order predictor and automatically generate the residual ("error") datasets for higher-order correction models. All commands assume you are in the repository root (see code in supplemental_material.zip).

1. **Log-centred warp & resampling for Model 1 data**
   - Place your pre-factorised first-order training CSVs into `testxiao/basic/model1/`.
   - Run:

   ```
   python testxiao/testxiao.py \
     --input_dir testxiao/basic/model1 \
     --output_dir testxiao/results \
     --warp center-log
   ```

   Intermediate warped traces appear in `testxiao/results/`.
   - Next, uniformly resample in $\tau$-space:

   ```
   python testxiao/uniform.py \
     --input_dir testxiao/results \
     --output_dir data/model1
   ```

   The resampled CSVs for Model 1 are now in `data/model1/`.

2. **Train the first-order model**
   - Launch training:

   ```
   python train_model1.py \
     --data_dir data/model1 \
     --save_path models/static_cond_model.pth
   ```

   Upon completion, the static first-order model is saved as `models/static_cond_model.pth`.

3. **Generate error datasets**
   - Execute the pipeline script to compute residuals:

   ```
   python pipeline_train_error_models.py \
     --model_path models/static_cond_model.pth \
     --input_dir data/model1 \
     --tmp_dir _tmp_ds
   ```

   This creates error-of-order-2 and error-of-order-3 traces in `_tmp_ds/`.
   - Copy the intermediate error files into the next data folder:

   ```
   cp _tmp_ds/error_*.csv data/error/
   ```

4. **Train the error models**
   - Finally, train the error prediction networks:

   ```
   python combine_and_train_error_model.py \
     --data_dir data/error \
     --save_prefix models/model_error
   ```

   This produces `models/model_error_2.pth` and `models/model_error_3.pth`, which predict the 2nd- and 3rd-order residuals respectively.

All script invocations include detailed usage notes in their headers. For full examples and parameter options, consult the source files in the `testxiao/` and root directories of our code in supplemental_material.zip

### D.3 Performance Evaluation

Once the static and error models have been trained (`models/static_cond_model.pth` and `models/model_error_n.pth`), we run the recursive inference script to generate test predictions and metrics. You may adjust inference parameters (e.g. batch size, lookback window) via command-line flags.

```
python test_recursive_inference.py \
  --data_dir     data/order1/test   \
  --static_model models/static_cond_model.pth \
  --err_model    models/model_error_n.pth    \
  --out_dir      results
```

The script outputs:

- Predicted vs. true response CSVs in `results/`

- Summary ($R^2$) printed to console and saved under `results/metrics/`

- Plots of step-response comparisons in `results/plots/`

## E  200-Order Extrapolation for Dynamic Signoff

To address the dynamic signoff mode, we conducted high-order experiments up to 200 poles. We trained residual-correction modules on systems with no more than 15 poles and evaluated on a 200-pole network. The model achieved a coefficient of determination of $R^2 = 0.9838$, indicating that our approach scales to orders required for full dynamic signoff with a still manageable training cost (approximately 5–6 hours on a single RTX 4090).

### E.1  Experimental Setup

We first generated training data for systems of order $\leq 15$ using standard *HSPICE* simulations. Residual-correction modules contained approximately 100–160K parameters and were trained with mixed precision on a single NVIDIA RTX 4090 GPU. Hyperparameters followed the settings used in lower-order experiments (optimizer, learning rate schedule, batch size), with early stopping based on validation $R^2$. The 200-pole evaluation network was held out from training and tuned to match the transient signoff configuration (10 ns horizon, 1,000 samples).

### E.2  Case Studies

We report aggregate metrics and qualitative comparisons against *HSPICE* on the 200-pole network. The model attains $R^2 = 0.9838$ with low mean absolute error and well-behaved residuals across the transient window. Visual overlays show close alignment at both fast and slow time scales, with the largest errors concentrated near rapid slope changes.

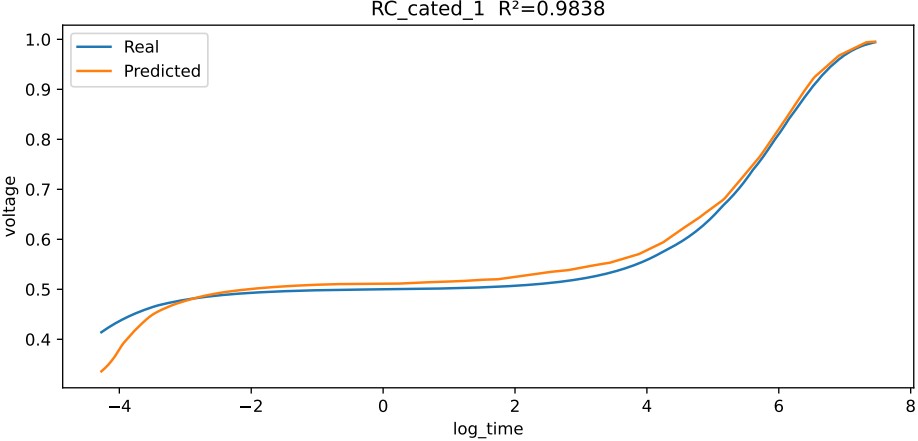

Figure 10: 200 order case study

### E.3 Discussion

These results demonstrate that the residual-correction design generalizes beyond the training order range, providing stable extrapolation to 200 poles. While inference remains fast and memory-efficient, training cost scales primarily with data volume and sequence length; the reported 5–6 hours is a one-time training cost and remains practical for signoff contexts. Remaining gaps are localized to high-curvature regions, suggesting that targeted augmentation (e.g., emphasizing rapid transients) and modest architecture scaling could further close the accuracy margin without compromising the observed efficiency benefits.

