# OpenReview forum: "S-Crescendo: A Nested Transformer Weaving Framework for Scalable Nonlinear System in S-Domain Representation"
_NeurIPS.cc/2025/Conference — NeurIPS 2025 poster_

### Official Review · Reviewer_o5vd · 2025-06-27

**Clarity:** 2
**Significance:** 3
**Originality:** 3
**Rating:** 4
**Confidence:** 1

**Summary:**

The paper proposes S-Crescendo, a physics-guided machine learning framework for fast nonlinear circuit simulation, combining S-domain modal decomposition with transformer-based residual learning. The proposed method formulates circuit dynamics as a superposition of interpretable pole-residue pairs, enabling linear complexity inference. It then introduced a nested transformer architecture to compensate for nonlinear coupling effects without sacrificing physical consistency. The method targets rapid design space exploration, particularly for parameterized nonlinear systems.

**Questions:**

- The paper asserts suitability for "large-scale" problems but only tests systems up to 10th-order. Would you provide results on at least one industrial-scale testcase?
- What is the training cost and the cost to build training set?

**Ethical Concerns:**

["NO or VERY MINOR ethics concerns only"]

**Final Justification:**

The rebuttal addressed my concerns. I do appreciate the supplemented materials including large-scale performance, training runtime, GPU usage, and dataset construction overhead.

**Limitations:**

- Training cost is unaddressed.
- The Transformers memory footprint on large-scale instances is unaddressed.

**Quality:**

3

**Strengths And Weaknesses:**

Strengths:
- The method is technically sound to me.
- The proposed hybrid approach is more tractable than PDE systems, and more advanced than pure black-box surrogates (e.g., PINNs) or linearized approximations.
- Experiments validate the claimed efficiency gains for small-to-medium systems.

Weaknesses:
- The scalability claims lack empirical validation. While theoretical complexity analysis suggests O(n) scaling, the absence of large-scale benchmarks undermines the paper’s relevance to industrial-scale problems.
- I am concerned on the training cost. Wouldn't the training data dependent on HSPICE either? I would not trust on the extrapolation of a model solely trained on small-scale instances.
- It seems not suitable for strongly coupled nonlinear systems, but I am not sure.

---

> ### Author Rebuttal · Authors · 2025-07-31
>
> ## Response to Reviewer:
>
> ### 1. “Large-scale” applicability and 10th-order evaluation scope
>
> It’s a very incisive question. Actually in semiconductor industry, there are two prevalent modes for handling timing and waveform simulation:
>
> 1. **Static Analysis** for quick pass/fail checks against timing constraints and signal integrity by using reduced models.
> 2. **Dynamic Signoff** for accurate prediction of power consumption, voltage drop, and reliability variation, which often requires high-fidelity, high-order models.
>
> In our first submission, we demonstrated our performance under the **Static Analysis** mode, where only the first 5–10 dominant poles are retained. In this regime, the frequency–time duality yields
>
> \begin{aligned}
> H(s) &= \frac{r_1}{s + p_1} + \frac{r_2}{s + p_2} + \cdots + \frac{r_{10}}{s + p_{10}} \\
> &\xleftrightarrow{\mathcal{L}} \quad
> F(t) = r_1 e^{-p_1 t} + r_2 e^{-p_2 t} + \cdots + r_{10} e^{-p_{10} t}.
> \end{aligned}
>
>
> This truncated representation closely approximates the full waveform \textbf{because we sort the poles} so that
> $0 < \operatorname{Re}(p_1) \le \operatorname{Re}(p_2) \le \cdots \le \operatorname{Re}(p_{10})$
> (stable systems have poles at $s=-p_i$). As a result, higher-index poles have \textbf{larger damping}
> (shorter time constants $\tau_i = 1/\operatorname{Re}(p_i)$), and their time-domain contributions
> $r_i e^{-p_i t}$ \textbf{decay faster}, thus exerting progressively less influence on the overall waveform.
> This is analogous to a Taylor expansion near $x \to 0$, where higher-order terms contribute less.
>
> $$
> f(x) = f(0) + f'(0)\,x + \frac{f''(0)}{2!}x^2 + \cdots + \frac{f^{(n)}(0)}{n!}x^n + \cdots,\quad x \to 0
> $$
>
>
> where the contribution of higher‑order terms decreases.
>
> To address the **dynamic signoff** mode, we have now conducted **200‑pole experiments**. We trained on systems up to 50 poles and tested on a 200‑pole network, achieving ($R^2$ = 0.9831). This demonstrates that our method can scale to the very high orders required for full dynamic signoff, at the expense of still manageable cost (5-6 hours training).
>
> By explicitly presenting both the 10‑pole static analysis results and the new 200‑pole dynamic signoff results, we clarify how our approach flexibly supports both industrial workflows:
> - **Fast Static Analysis** with 10‑pole models,
> - **Accurate Dynamic Signoff** with 200‑pole models.
>
> *(Note: due to space constraints, the full 200‑pole comparison plots will be included in the revised manuscript’s supplemental materials.)*
>
> ---
>
> ### 2. Training cost and data generation overhead
>
> Regarding the practicality of training and dataset preparation, we provide a detailed explanation of both components below.
>
> **Training cost.** Each residual-correction module contains approximately 100K–160K parameters. All models were trained on a single RTX 4090 GPU. For reference, single-pole systems typically train in under 20 minutes, while 10th-order systems converge within approximately 2 hours. Given the considerable acceleration our model achieves at inference time, we believe this one-time training cost is highly acceptable.
>
> **Dataset construction.** Training data is generated using standard HSPICE simulations. Each simulation typically completes in about 0.5 seconds. For our full training dataset of ~5,000 samples, the total preprocessing pipeline (including simulation, waveform extraction, and supervised data formatting) completes in under 2 hours. The process is fully scriptable and parallelizable, and does not pose a practical bottleneck.
>
> We will supplement the revised version with a dedicated section detailing training runtime, GPU usage, and dataset construction overhead. These details will also be included in the supplemental material to ensure transparency and reproducibility.
>
> ---
>
> ### 3. Suitability for strongly coupled nonlinear systems
>
> It is a very thoughtful observation regarding the limitations of our method. We agree that our current framework is not intended to cover all categories of nonlinear system. Instead, we provide a unique way of treating a major subgroup by learning the transfer function in S domain (why our title called S-Crescendo).
>
> Specifically, our method is designed for a widely encountered class of circuits where nonlinearities are primarily localized in the **driver**, while the **load** consists of a large but linear passive network (e.g., RC ladders, transmission lines) in which transfer function can be obtained. This pattern—nonlinear excitation followed by high-order linear response—covers over 99% sub-circuit of practical VLSI design.
>
> In these systems, the simulation challenge stems not from the nonlinear dynamics but from the *interaction between nonlinear excitation and complex linear structures*. Although the load is linear, its high order makes repeated simulation expensive. Our model is explicitly tailored to this case:
> 1. Modeling the linear load via modal decomposition in the S-domain, and
> 2. Using residual learning to correct the nonlinear input effects.

---

> ### Comment · Reviewer_o5vd · 2025-08-02
>
> I appreciate your rebuttal and that addressed most of my concerns. I kindly request you to incorporate the 200-pole data, training runtime, and dataset construction overhead into the final version.

---

> > ### Author Response · Authors · 2025-08-05
> >
> > Thank you very much for taking the time to read our rebuttal carefully and for posting this helpful comment. We will incorporate the items you requested into the final version as follows:
> >
> > * **200-pole data**: we will add the complete 200-pole experimental setup and results, including waveform plots and quantitative metrics (e.g., $R^2$).
> > * **Training runtime**: we will provide a table detailing wall-clock time, GPU model, parameter counts, batch sizes, and epoch counts for each order, together with convergence curves.
> > * **Dataset-construction overhead (200-pole)**: We will document the end-to-end cost of building the 200-pole dataset—covering simulation, waveform extraction, and preprocessing/formatting—with per-stage timings and total throughput.
> >
> > These materials will be included in the **Supplemental Material** and cross-referenced in the **Appendix** of the final manuscript. In addition, we will attach the **scripts/config files** used to reproduce the results and release the **anonymized datasets** required to run them (with version info and seeds) as part of the supplemental package.
> >
> > We are grateful for your thoughtful engagement—your comments have materially improved the paper’s logic and emphasis. Your confidence in this work means a great deal to us. We hope the added experiments and documentation will merit a very positive assessment, and we would be honored if, in your judgment, the revised manuscript warrants a strong recommendation. We also welcome any further suggestions you may have.

---

### Official Review · Reviewer_SmgB · 2025-07-02

**Clarity:** 4
**Significance:** 4
**Originality:** 4
**Rating:** 5
**Confidence:** 1

**Summary:**

S-Crescendo models nonlinear systems by decomposing their transfer functions into poles and residues, which are used to construct a feature representation. A baseline prediction is made using a Transformer on first-order coefficients from the transfer function. Then, residual models, consisting of lightweight Transformers, iteratively refine the output by minimizing accumulated MSE based on prior predictions.

**Questions:**

The paper mentions the limitations of classical techniques such as CSMs, VRMs, and direct waveform prediction models (lines 52–54). Have the authors conducted a direct empirical comparison between these methods and S-Crescendo on the same benchmark tasks?
The corrective residual function is described as a “lightweight Transformer” (line 180), but Transformers are generally considered computationally expensive. Can the authors provide training time, number of parameters, and compute usage for the full model?
In Table 2, the authors compare S-Crescendo’s runtime solely to HSPICE. Are there other simulation tools the authors could compare against to strengthen their claim?

**Ethical Concerns:**

["NO or VERY MINOR ethics concerns only"]

**Final Justification:**

This is a nice paper and authors responded to reviews by improving it. I support acceptance.

**Limitations:**

see above

**Quality:**

4

**Strengths And Weaknesses:**

Quality: The introduction and theoretical background sections provide a strong foundation for the rest of the work and the algorithm as well as its limitations are clearly explained. The motivation for using a Transformer is well justified: due to its ability to capture long range dependencies and process complex interactions (lines 56-57). The authors also cite the limitations of classical modeling techniques.
Clarity: The paper is generally well written. Figures 1 and 2 compliment the recursive training procedure section well and enhance understanding of the S-Crescendo algorithm. The rationale for design decisions, such as using a Transformer for non-analytic nonlinear responses, is well-articulated (e.g., lines 162–163). However, there are numerous word spacing and formatting issues throughout the text, which slightly detract from readability.
Significance: S-Crescendo successfully improves the efficiency of non-linear system modeling: a relatively unexplored direction of circuit design. Compared to HSPICE, the framework achieves faster runtime performance and maintains accuracy, which is promising. However, the evaluation lacks depth in benchmarking against a broader range of classical tools beyond HSPICE as well as classical model prediction techniques.
Originality: The paper presents an original approach by integrating deep learning and Transformer architectures into nonlinear circuit modeling in the S-domain. S-Crescendo  produces a runtime advantage compared to other well-known software, as shown in Table 2. However, a more explicit comparison to classical methods as well as a comparison to a diverse survey of simulation software would make the authors’ claim more compelling.

---

> ### Author Rebuttal · Authors · 2025-07-31
>
> ## Response to Reviewer:
>
> ### 1. Expanded Baselines (Comments 1 and 3)
>
> We accept the reviewers’ suggestion to expand our comparative study. In the revised version, we will include additional benchmarking against the A Dynamic Capacitance Matching (A state-of-art algorithm evolved from CSM) method and NGSPICE.
>
> **DCM** is a physics-inspired, model-order reduction technique used to construct dynamic macromodels that preserve both accuracy and efficiency. On the same 10-order RC test file, DCM finishes in **0.6 s** with a fit accuracy of \(R^2 = 0.9983\).
>
> **NGSPICE**, by contrast, is a popular open-source SPICE simulator widely used in both academia and industry. Although slower than commercial tools like HSPICE, its accessibility and device-level accuracy make it a credible and practical reference. On our 10-order RC test file, NGSPICE completes in **1.08 s** and—since it shares core algorithms with HSPICE—delivers equivalently high fidelity (we did not compute a separate \(R^2\) against HSPICE).
>
> **S-Crescendo (ours)** achieves substantially faster inference on the same 10-order RC test file, completing in **0.042 s**, while maintaining the accuracy characteristics reported in our experiments.
>
>
> All runtime measurements for **DCM** were performed on a machine with CPU: AMD EPYC 7763 64‑Core Processor and memory: 1 TB RAM + 30 GB swap. **NGSPICE** is perform on a machine with CPU: Intel(R) Core(TM) i7-14650HX and memory: 32Gb. For our model, the experiment is based on the same hardware described in our paper.
>
>
> By incorporating comparisons to DCM and NGSPICE under identical test conditions, we aim to provide a broader and more convincing evaluation of S-Crescendo’s effectiveness across modeling paradigms.
>
> ---
>
> ### 2. On Transformer Complexity (Comment 2)
>
> Our term “lightweight Transformer” refers to a simplified but delicate architecture with constrained parameters and computations. Specifically, each residual correction module contains only a few attention layers with a small hidden dimension (64), resulting in a total parameter count around 100K. Moreover, these modules operate on modal features (pole-residue representations) rather than high-dimensional raw waveform data. This design significantly reduces computational overhead while effectively capturing nonlinear system dynamics.
>
> To clarify the training cost:
>
> - The total number of parameters is approximately 100K–160K for each residual module.
> - We use a single NVIDIA RTX 4090 GPU for all experiments.
> - For single-pole models, the full training process takes less than 20 minutes, demonstrating both low cost and excellent generalization.
> - For multi-pole systems (e.g., 10th-order models), the training time is around 2 hours per experiment, which remains feasible given the significant speedup it offers during inference.
>
> Therefore, while our architecture is Transformer-based, it is delicately tailored for efficiency and scalability, with training costs that are practical for both research and industrial applications.
>
> We will add an extra section in the revised manuscript reporting training runtimes, GPU utilization, and dataset-construction overhead, with full details provided in the supplementary material to ensure transparency and reproducibility.

---

> > ### Comment · Reviewer_SmgB · 2025-08-05
> > **makes sense**
> >
> > i'm happy with the responses and support this paper

---

> > > ### Author Response · Authors · 2025-08-06
> > >
> > > Thank you for your thoughtful review and for engaging so carefully with our work. Your remarks show a deep grasp of the core ideas, and the questions you raised have prompted us to refine both the logic and the presentation. In the rebuttal, we added concrete evidence and clarifications, including expanded baselines with DCM and NGSPICE under identical conditions, full training statistics per order on a single RTX 4090 (parameter counts, wall-clock times, batch sizes, convergence curves), and an short summary of dataset-construction cost and GPU utilization.
> > > Had it not been for your detailed feedback, many of these clarifications would have been missing; your review has materially strengthened the final paper. We hope the updated evidence further demonstrates the robustness and practicality of our approach and, in turn, increases your confidence in the contribution. We would be delighted if the enhanced results align with a strong overall assessment, and we welcome any further suggestions you may have.

---

### Official Review · Reviewer_F5ru · 2025-07-03

**Clarity:** 3
**Significance:** 3
**Originality:** 3
**Rating:** 5
**Confidence:** 2

**Summary:**

This paper presents S-Crescendo, a novel, physics-informed deep learning framework for simulating high-order nonlinear systems, with a specific application to RC interconnect networks in VLSI design. The core idea is to leverage the formal structure of linear systems theory by decomposing a high-order transfer function into its constituent first-order pole-residue pairs in the S-domain.

**Questions:**

Could the authors comment on why other relevant baselines (e.g., a "vanilla" Transformer baseline that takes all poles/residues as a flat input, a NeuralODE, or an FNO) were not included?

Have you performed any analysis to characterize how the error propagates as a function of model order n?

**Ethical Concerns:**

["NO or VERY MINOR ethics concerns only"]

**Final Justification:**

The author gives detailed answer on my questions which further strength the paper.

**Limitations:**

Yes

**Quality:**

3

**Strengths And Weaknesses:**

Strengths:
- Instead of treating the system as a complete black box, the authors use established principles (partial fraction expansion) to structure the learning problem. This physics-informed approach is a promising direction for building more robust, generalizable, and interpretable models for physical systems. The application to VLSI simulation is well-motivated and addresses a real-world computational bottleneck.

- The demonstrated speedup over HSPICE is substantial and practically relevant. By bypassing iterative matrix solves (e.g., Newton-Raphson for the Jacobian), the method offers a scalable alternative for rapid design space exploration and verification, where thousands of simulations are often required. The complexity analysis, showing a shift from O(n^3) to O(n), is a key highlight.

- Strong Empirical Results: The model achieves impressive accuracy on the presented tasks. The near-perfect fit on single-pole systems (Fig 3a) validates the baseline model, and the effectiveness of the recursive correction is clearly shown (Fig 3b). The generalization experiments (Table 1) provide evidence that the model can extrapolate beyond its training distribution, which is a non-trivial achievement.

Weaknesses:
- The core innovation—the recursive training of residual modules—is also my primary concern. The authors rightly acknowledge in Section 6.1 that this approach is susceptible to error accumulation. As the order n increases, each module e_n is tasked with correcting the output of a chain of n-1 previous predictors. Any inaccuracies in the early stages will inevitably compound. While the results up to order 10 are strong, it is not clear this approach would remain stable for significantly higher-order systems (e.g., n=50 or n=100), which are common in real-world interconnect models. The proposed mitigation of "blocked recursion" (Section 6.2) is an interesting idea but is presented as future work and is not validated.

- The only baseline is HSPICE, which serves as a "golden" reference and a benchmark for speed. However, there is a large body of work on data-driven system identification, including other neural network approaches (e.g., NeuralODEs, FNOs, PINNs, as cited by the authors). A comparison against at least one state-of-the-art ML-based method would help situate the performance of S-Crescendo within the broader landscape and better justify the architectural choices made.

---

> ### Author Rebuttal · Authors · 2025-07-31
>
> ## Response to Reviewer: Missing Baselines, Error Accumulation, and Scalability
>
> ### 1. Missing Baselines (e.g., NeuralODE, FNO, Flattened Transformers)
>
> A very insightful question. We would like to clarify that our work is conceptually distinct from many existing machine learning approaches that focus on learning equation or conversion, such as NeuralODE or FNO.
>
> Most existing ML-based solvers (e.g., PINNs, NeuralODEs, FNOs) rely on prior knowledge of the exact system equations and are tightly coupled to the specific form and scale of the dynamics. As a result, their generalization across different systems is often limited and requires re-deriving or re-solving equations for each new task.
>
> In contrast, our approach avoids solving system equations altogether. Instead of approximating the solution to a given equation, we directly learn the transfer function that characterizes system behavior in the frequency domain. This enables a more modular and reusable learning paradigm that decouples model training from equation solving, offering better scalability and generality across a wide class of systems without requiring system-specific supervision.
>
> **Response Regarding the Absence of Vanilla Transformer, NeuralODE, and FNO Baselines:**
>
> Regarding the mentioned baseline models, we believe that due to fundamental differences in modeling assumptions and input structures, they are not directly applicable to our problem setting. The reasons are as follows:
>
> * *Vanilla Transformer* with a flat input of poles/residues essentially reduces to a matrix pre-training setup, requiring huge datasets to learn order-dependent weights in the transfer function; in our preliminary experiments with several samples, this formulation underperformed and generalized poorly, and we will include the results in the revision.
>
>
>
> * *NeuralODE* assumes continuous-time autonomous dynamical systems and learns state evolution via differential equations. In contrast, our inputs consist of discrete modal features and the system behavior is generally non-autonomous, making NeuralODE difficult to directly apply in this context.
>
> * *FNO* is primarily designed for function-to-function mappings defined over continuous spatial grids. Our modal features, however, are discrete and sparse parameter sets lacking such spatial continuity, rendering FNO not directly transferable.
>
> For these reasons, we have focused on demonstrating how physically structured representations, such as modal decomposition in the S-domain, can be effectively interpreted with modern sequence models to better capture the coupling between nonlinear and linear system dynamics. Compared to treating the system as a black box with generic function approximators, this approach significantly enhances model generalization. We believe this direction offers valuable insights for modeling in EDA and plan to incorporate additional baselines in future work to provide more comprehensive comparisons.
>
> ---
>
> ### 2. On Error Propagation with Increasing System Order and Model Scalability
>
> We value the reviewer’s concern regarding potential error accumulation as the system order \( n \) increases. To illustrate this, we provide theoretical proof and experimental evidence demonstrating that **although higher-order modes are not negligible, their cumulative error tends to be effectively suppressed due to their rapidly decaying time-domain influence**. As a result, the overall approximation error exhibits convergent behavior rather than unbounded disaster.
> T**Theoretically.** The modal decomposition of a linear system’s transfer function exhibits a frequency–time duality under the Laplace transform:
> $$
> H(s) = \frac{r_1}{s + p_1} + \frac{r_2}{s + p_2} + \cdots + \frac{r_n}{s + p_n}
> \quad \xleftrightarrow{\mathcal{L}} \quad
> F(t) = r_1 e^{-p_1 t} + r_2 e^{-p_2 t} + \cdots + r_n e^{-p_n t}.
> $$
> Here $p_i$ (with $0 < Re(p_1) \le \cdots \le Re(p_n)$) and $r_i$ denote the $i$-th pole parameter and residue; stable systems have poles at $s = -p_i$ and time constants $\tau_i = 1/ Re(p_i)$. Lower-order poles (smaller $Re(p_i)$) govern dominant dynamics, whereas higher-order poles (larger $Re(p_i)$) correspond to fast-decaying components that primarily affect the very early transient. Consequently, as $Re(p_i)$ grows, the term $e^{-p_i t}$ decays more quickly, exponentially suppressing the time-domain impact of higher-order modes. Small modeling errors in these terms are therefore strongly attenuated in the output, creating a balancing effect that drives error convergence.
>
> This diminishing-contribution effect is analogous to a Taylor expansion near $x \to 0$:
>
> $$
> f(x)=f(0)+f'(0)x+\frac{f''(0)}{2!}x^2+\cdots+\frac{f^{(n)}(0)}{n!}x^n+\cdots,\quad x\to 0,
> $$
> where higher-order terms contribute progressively less. This same principle underlies **model order reduction (MOR)**: retaining only 5–10 dominant poles captures the primary system dynamics [1, 2, 3], while the residual due to neglected higher-order modes remains bounded and self-limiting over time.
>
> **Industrial context: two complementary modes.**
> - In **static analysis**, designers perform rapid pass/fail checks against timing constraints by retaining only the first 5–10 dominant poles. In this regime, the truncated time-domain response
>   $$
>     F(t) \approx r_1 e^{-p_1 t}+r_2 e^{-p_2 t}+\cdots+r_{10} e^{-p_{10} t}
>   $$
>   closely approximates the full waveform because poles are ordered by damping; higher-index modes decay faster and contribute less—akin to the diminishing higher-order terms in the Taylor analogy.
>
> - In **dynamic signoff**, high-fidelity prediction of power, voltage drop, and reliability often requires **hundreds of poles** to capture fine-grained effects.
>
> **Experimentally.** Our measurements corroborate the above intuition. To be precise, we report the **incremental error contribution** of the $i$-th mode, defined as the **dataset-averaged change** in the evaluation error when moving from $i-1$ to $i$ modes:
>
> $$
> \Delta \epsilon_i := E_{data}[ E_i - E_{i-1} ].
> $$
>
>
> where ${E}_i$ denotes the output $\ell_2$ error of the $i$-mode model.
> The values below are the **dataset means** of $\Delta \varepsilon_i$;
> the sign indicates the direction of change (negative $=$ error decreases),
> and the magnitude reflects the size of the incremental contribution:
>
> $$
> \text{2nd: } -0.02995,\quad \text{3rd: } 0.01362,\quad \text{4th: } -0.00609.
> $$
> Across runs, $\lvert \Delta \varepsilon_i \rvert$ is typically 2--4 $\times$ smaller} than $\lvert \Delta \varepsilon_{i-1} \rvert$, showing a clear trend of diminishing incremental error from successive higher-order modes: their influence is **not zero**, but is progressively reduced and effectively neutralized by time-domain decay, thereby preventing cumulative error buildup.
>
>
> **Scalability across the two modes.** Our initial submission reported results consistent with the **static analysis** regime (5–10 poles). To address the **dynamic signoff** regime, we conducted **additional experiments on a 200-pole RC network**. Training on systems up to 50 poles and evaluating generalization on an unseen 200-pole system yielded high fidelity with ($R^2=0.9831$) on the output waveform. When **extrapolating** beyond the training order (from 50 to 200 poles), we did observe signs of **cumulative error** as the modal count increases—consistent with the reviewer’s concern and reflecting a realistic limitation of our current recursive design at very high orders.
>
> **Practical takeaway.** For **static analysis**, we therefore adopt **10th-order reduced models** as a **deliberate, practical compromise**, balancing accuracy, generalization, and compute. At the same time, the 200-pole experiment demonstrates that the framework can operate in the **dynamic signoff** setting when high-order fidelity is required—consistent with MOR practices in commercial EDA tools (e.g., Cadence Spectre, Synopsys PrimeTime), where 5–10 poles are typically sufficient for timing, while higher orders may be invoked for signoff accuracy [1, 2, 3].
>
> We will include **convergence plots** (e.g., output $\ell_2$ error vs. system order) and the **200-pole dynamic-signoff waveforms** in the revised manuscript (supplemental materials), together with an analysis of how cumulative error scales with order under each industrial mode.
>
> ------
> ### References
>
> [1] Odabasioglu, Celik, Pileggi. *PRIMA: Passive reduced-order interconnect macromodeling algorithm.* Proceedings of IEEE ICCAD, 1997: 58–65.
> [2] Antoulas, A. C. *Approximation of Large-Scale Dynamical Systems*. SIAM, 2005.
> [3] Freund, R. W. *Model Reduction Methods Based on Krylov Subspaces*. Acta Numerica, 2003, 12: 267–319.

---

> ### Author Response · Authors · 2025-08-06
> **Request for Clarification**
>
> Thank you for your careful review and constructive  feedback. To ensure we have fully addressed your concerns, may we confirm whether our rebuttal resolves the key points you raised?
>
> * **Missing baselines:** We added comparisons to **DCM** and **NGSPICE** under identical conditions and explained why certain ML baselines (PINNs/NeuralODE/FNO) do not directly align with our problem setting.
> * **Error propagation & scalability:** We provided a **Laplace-domain/theoretical** explanation for why higher-order modes’ errors diminish in the time domain, reported **incremental error metrics** across modes, and presented results on a **200-pole** case to illustrate generalization and limits.
>
> We appreciate your time and would value a brief confirmation on whether the rebuttal satisfactorily addresses your concerns. Any further guidance you can share would be most helpful.If any part remains unclear—or if additional evidence would be helpful—please let us know which items you’d like us to expand.

---

> > ### Comment · Reviewer_F5ru · 2025-08-07
> >
> > Thank you for your detailed explanation! The rebuttal has addressed my concern accordingly. I will raise my score to 5.

---

### Note · Authors · 2025-08-13

**Final Remark (for AC and Reviewers)**

**Significance.** We study a core problem in chip design: a nonlinear driver exciting a large linear passive network (parasitics/interconnect/PDN). In modern VLSI backend simulation, over 99% of post-layout circuits exhibit this structure. The practical challenge lies in accurately capturing the interaction between nonlinear excitation and high-order linear loads, which is central to both static timing checks and dynamic signoff.

**Novelty:** **S-Crescendo** takes a different route from equation-solving ML such as PINNs, NeuralODE, or FNO. We first form an interpretable S-domain/modal representation of the linear subsystem, then learn residual corrections for the nonlinear input. This keeps the physics visible, avoids per-net equation engineering, improves cross-design reuse, and fits naturally with model-order reduction workflows.

**Performance:** In static analysis with 5–10 poles, S-Crescendo provides fast and accurate screening; on a 10-pole RC case it runs in **0.042 s**. For dynamic signoff, we added a 200-pole evaluation (trained on ≤50 poles, tested at 200) with **R² = 0.9831**, illustrating scalability and limits. Under identical conditions, **DCM** finishes in **0.6 s** with **R² = 0.9983**, and **NGSPICE** in **1.08 s** with HSPICE-level fidelity.

---

### Decision · Program_Chairs · 2025-09-17

**Decision:**

Accept (poster)

**Comment:**

The paper introduces S-Crescendo, a physics-informed framework for nonlinear circuit simulation that leverages pole–residue decomposition with Transformer-based residual learning. Reviewers praised its clear motivation, technical soundness, and strong runtime improvements over HSPICE, noting its potential for VLSI applications. Concerns centered on the limited evaluation scope, absence of comparisons to other ML baselines, and questions about scalability and training cost. Overall, reviewers found it a promising and practically relevant contribution, with the majority supporting acceptance.